# Using Event-Related Potentials to Evidence the Visual and Semantic Impact: A Pilot Study with N400 Effect and Food Packaging

**DOI:** 10.3390/foods13121876

**Published:** 2024-06-14

**Authors:** Juan-Carlos Rojas, Manuel Contero, Margarita Vergara, Juan Luis Higuera-Trujillo

**Affiliations:** 1Escuela de Arquitectura, Arte y Diseño, Tecnologico de Monterrey, Monterrey 64700, Mexico; 2Instituto Universitario de Investigación en Tecnología Centrada en el Ser Humano, 46022 Valencia, Spain; mcontero@upv.es; 3Departamento de Ingeniería Mecánica y Construcción, Universitat Jaume I, 12071 Castelló de la Plana, Spain; vergara@uji.es; 4Departamento de Ingeniería Mecánica y Diseño Industrial, Universidad de Cádiz, 11510 Puerto Real, Spain; juanluis.higuera@uca.es

**Keywords:** packaging, N400, event-related potential (ERP), design artwork

## Abstract

Packaging design is pivotal in motivating consumer decisions, as a key communication tool from creation to purchase. Currently, the interpretation and evaluation of packaging’s impact are shifting toward non-traditional methods. This pilot study evaluated the packaging perception of York Ham and Turkey Breast products. The event-related potential (ERP) technique, the methodology priming words (positive and negative), and target images (original and modified packaging) were applied. A total of 23 participants were sampled using a 32-channels scalp elastic electrode cap and viewed 200 trials of word–image matching. Participants responded whether the images and adjectives matched or not, using the two groups of images. The results demonstrate an N400 effect in the parietal area. This region was observed to show evidence of cognitive processing related to congruency or incongruency, by contrasting the priming and target of this study. The evaluation positioned the York Ham packaging as the best rated. The findings show a relevant contribution to ERPs and research related to the food packaging perception.

## 1. Introduction

Packaging design has become a crucial element in the entire product purchasing process; the visual appeal of packaging artwork motivates consumers to process information about the whole product message [1]. Packaging establishes an important form of communication from creative creation to consumption, liking, and willingness to purchase [2]. Packaging brings those first touching points to be made quickly and with minimal effort for purchase decisions [3,4], and through different design characteristics, consumers can make the most convenient decision [5]. However, the complexity or simplicity of the message sent by the packaging is something that must be studied deeply. It is well known that consumers view, read, and process information as a coherent or congruent message, in which each design element conveys a particular component to the consumer for choice, preference, and purchase [6,7]. There are numerous approaches to understanding the factors that contribute to the success of a package and how consumers perceive it. Packaging should stand out from the sideboard or shelf and scream out to be noticed [8]. In this sense, attention and emotion evoked by artwork play a crucial role in the effectiveness of packaging design, to facilitate visual information processing for consumer decision-making [9,10,11,12]. The visual information of a package is mainly constituted by elements such as color, image, logos, and typeface [4,13]; even so, this visual conjunction of elements in packaging composes a message planned by a few people and which has nothing to do with the consumer’s final perception. The creative processes that lead to packaging development can have their origin in several market strategies [14] or are driven by empirical design methodologies [15], which are subsequently assessed by the target audience. Consequently, packaging design must consider and balance diverse elements while the essential message is processing, and principally, the viewer has to quickly decipher what they look at [16,17]. In light of the aforementioned, we shall embark upon an in-depth exploration of the numerous studies surrounding the perception and consequential impact of packaging design.

### 1.1. Seeking Objective Evaluation of the Impact of Packaging Design

One of the objectives of this research is to address a question related to the perception of food packaging, especially from an objective perspective on the impact of the visual-written message and its comprehension. Understanding the link between objective perception and visual impact will help to design better packaging. Drawing upon these two premises, this study attempts to inquire into how we can measure this objective perception, in a constantly changing context between consumers, strategies, marketing goals, and a large number of options that provide value to consumers [18]. More specifically, we will explore how packaging can convey information related to its appearance and the product it contains, through its visual/stylistic and verbal characteristics, from a proposal designed by creative or strategic processes [17,19]. Historically, we have come a long way in the evaluation of packaging; however, the further objective evaluation of packaging’s impact is a milestone today. The traditional measurement methods have provided us with insights into how the visual elements of packaging can convey both denoted and connoted information to consumers [20]. In fact, textual information displayed has an important effect on consumer’s perception, principally the expectations of a product [8], adding up the impact of the images to generate more engaging and vivid information accompanying text information [20,21] and complementing with other types of labels like promotion or nutrition-specific labels [22,23]. One of the main drawbacks in impact interpretation and assessment is that traditional methods are limited to providing relevant but biased information. Multiple facets of product design, particularly in the domain of packaging, have been gradually embracing innovative evaluation techniques. Notably, this shift has witnessed a significant acceleration in recent years [24]. Several studies, primarily focused on neurosciences, have demonstrated the effectiveness of neuro-techniques in analyzing the perception of the aesthetics, brands, packaging, and pricing of products. This extends to applications not only in the commercial domain but also in therapeutic or recreational contexts [25,26,27,28,29,30,31,32,33,34]. This emerging trend aligns with a broader movement in design research that seeks to integrate technological advances, sociocultural considerations, economic factors, and mixed methodological approaches into the foundations of contemporary design [35,36,37,38,39].

### 1.2. Understanding Cognitive Processes in Consumer Perception through Event-Related Potentials and N400 Effects

In technological shifts and embodiment, electroencephalography (EEG) stands as one of the most widely employed methods for the observation of brain activity and cognitive processes for positive effects [40,41]. Particularly, this technology allows the performance of a technique known as event-related potentials (ERPs), which provides a non-invasive means of collecting data that can reflect participants’ psychological activity and responses [42,43]. ERPs’ nature has shown that the changes observed in the brain electrophysiological signal are useful for examining the perceptual and cognitive processes in product assessments [44]. ERPs make possible the recording of electrical activity during an activity or experience, particularly in a signal observation window known as N400. The term “N400” originates from the peak latency observed in the first studies [45] showing the relationship between brain activity and type localization, which was 400 ms [46]. However, this activity is observed as a negative deviation in the observation window, just after the presentation of some stimulus in the frontal and central areas of the brain [47,48]. This peculiarity drew attention from its early evidence owing to an effect observed in tasks involving a semantic processing load and the congruence among the stimuli used [45].

Another objective of this research is to ensure the observation of N400 effects, as some research has favored a specific experimental arrangement that guarantees the making of congruent or incongruent associations between the observed stimuli by using the semantic priming technique. As relevant references, we have observed in several studies how the lexical-semantic context of congruency or incongruence with stimuli has triggered the N400 effect with ERPs [49,50,51,52,53,54,55,56]. More recently, our attention has been focused on the fact that the N400 effect is not limited only to lexical stimuli with associated semantic context attributes [57], but it also extends to direct associations between visually ordered stimuli. Several studies have demonstrated how the semantic context evoked by nonverbal stimuli, i.e., graphic elements, triggers the N400 effect. Examples of these have been images–text [58], familiar–unfamiliar faces [59], drawing–drawing [60], sentences–sentences [61], and pictures–pictures [62]. In all these cases, we can see an N400 effect evoked by the priming technique, where one chooses a prime (text or image) and a target (text–image) that can be congruent or incongruent. An example [63] of congruence is the image of a cat before the image of a dog; for incongruence, it is the image of a knife before the image of a dog. With all this evidence, the N400 is a relevant indicator for the understanding of cognitive processing that we should take into consideration.

This research focuses on the importance of the priming effect as the word and the target as the image. Considering similar work that uses semantic and visual elements to elicit the N400 effect, we find that congruency is the first effect that can be observed in the stimulus contrast [64,65]. Congruency as a trigger of the N400 effect has been observed in research for visual stimuli such as geometric drawings, nature, or everyday images [66,67,68,69]. On the opposite side, incongruence has also been observed in research using stimuli to generate a contrast between adjectives and nouns [70,71], or to enhance the effect with targets with normal or distorted images [72] or the compression or relationship between a priming word and a graphic element [73]. In recent years, N400 effect observation by ERPs has been applied in areas related to consumer, user, and product aspects, as this has also impacted the emotional domain related to the stimuli presented. ERPs have consolidated a method to obtain the effects on the congruence or incongruence of stimuli in multiple contexts. The first element to highlight is the priming or prime of semantically and/or emotionally charged words (positive, neutral, and negative) [74,75,76,77,78]. The second element to highlight is the targets, which are associated or decontextualized images that evoke the ERPs [74,75,76,77,78].

Recent advances in ERP applications have led to renewed interest in its relevance to decision-making, cognitive processing, and the emotional perception of brands, products, and aesthetic-visual attributes. Examples can be found of how product exposure or brand-relatedness [79,80,81] modulate the N400 effect and also how a product and its attributes such as color, material, or lighting paired with descriptive information such as attributes or descriptors trigger congruence or incongruence effects [82,83,84,85]. These examples have also shown emotional elements implied by the use of priming or visual targets that show products [86,87,88]. Furthermore, recent reference research has evidenced potency as a catalyst of the congruence effect, mainly incongruence, in the relationship between a product, brand, and object characteristics or tributes [81,82,88]. This relationship was found for the N400 effect where color, materials, or similar can provide a significant incongruence effect in poorly designed or poorly displayed items [47,84,85].

Hence, our research will take the aspects described above in the review of similar research. Our hypothesis postulates that the N400 effect will be observed in the contrast between semantic priming related to the product attributes on the packaging and the visual targets of the selected packaging. In this study, we will record the congruence or incongruence presented by the comparison between words and images.

## 2. Materials and Methods

### 2.1. Participants

Twenty-eight undergraduate students participated in the study. After the EEG/ERP process, only 23 students’ data (74% male), with a mean age of 23.3 years (SD = 2.5 years), were suitable for use because of the decrease in signal quality with hair thickness. The final sample has an average sample number above the mean value typically used in this type of study. Šoškić et al. [89] analyzed 132 N400 studies, observing a mean N of 20.2 participants. All the participants were native Spanish, right-handed, and had normal or corrected vision. This research complied with Tecnológico de Monterrey Code of Ethics and was approved by the Ethics Committee as a non-invasive experiment for participants. Informed consent was obtained from each participant.

### 2.2. Stimuli

This pilot study uses the priming and targeting methodology to provoke the N400 effect. A series of Spanish-branded deli meat packaging samples were taken as targets. The selection decision was based on a better understanding of the visual elements [90] that make up the packaging of York Ham and Turkey Breast (see Figure 1). Following the methodology, we used semantic priming, words that are obtained from the perception of attributes and characteristics related to the brands and the products contained in the packaging [90]. Table 1 shows the prime words used in Spanish and their translation into English. Considering the nature of semantic priming, five positive adjectives were selected. Negative adjectives were also adopted as the closest antonyms. The study was conducted in Spanish.

The experimental setup was performed as described below. Two groups were set up: Group 1 with 10 images of the original packaging; and Group 2 with 10 images of modified versions. The images in Group 1 (original packaging) contain 10 images: 5 images for “York Ham” packaging and 5 images for “Turkey Breast” packaging (see Figure 1). The images in Group 2 (modified packaging) were all for “York Ham”, edited to add or remove the euro sticker (1€) and interchange brand logos (see Figure 2). Note that the packaging of both products is quite similar and some of them present the promotion of a euro sticker. All targets’ stimuli (740 × 500 pixels) were shown centered on the screen (1090 × 1080 pixels).

### 2.3. Procedure

The study was performed in a dimly lit laboratory room. Participants were sat in front of a computer screen at a 65–70 cm distance. Each participant was equipped with two elements: a 32-channels scalp elastic electrode cap and an EMG sensor. To prevent blinking and eye movement recording, the EMG sensor was placed at 1.5 cm near to the right eye and eyebrow. Participants were instructed not to move their heads and legs during the experiment. The study was programmed and presented using Neurolab^®^ (version 1.2, Bitbrain Technologies, 50006 Zaragoza, Spain). During the development of the test, participants performed a match task using two response buttons to express their personal perception of priming and the target combination for all combinations of the 10 words and 20 images. A total of 200 trials divided into two blocks were presented. One block was composed of positive valence words with 20 target images corresponding to original packaging images and modified packaging images. The other block was composed of negative valence words with 20 target images. The priming–target combinations appeared randomly from one of the two groups only once during the experiment. Each trial (see Figure 3) began with a fixation cross-point at the center of the screen for 400 ms. Afterward, a positive or negative adjective (priming) was displayed for 700 ms, followed by a neutral screen (white color) for 200 ms. Finally, the target image was displayed for 1700 ms, followed by a question about the right match between the priming and the target. To finish the trial, participants must press a “yes” or “no” button as soon as they make their match. Subsequent trials started immediately after the participant entered their response.

### 2.4. Electrophysiological Recording and Analysis

The EEG signals were continuously recorded using Neurolab^®^ with a REFA digital amplifier system (TMSi company, 7575 EJ Oldenzaal, The Netherlands) at a sampling rate of 256 Hz with Ag/AgCl water-based electrodes, a wet band served as the ground, and impedances were kept below 5 kΩ. The EMG sensor recorded the vertical electro-oculography (EOG) [76]. The recording was collected based on a common reference. The continuous EEG signals were filtered offline using a 30 Hz low-pass filter and segmented in epochs of 200 ms before and 800 ms after target onset. Following this, an Independent Component Analysis (ICA) applying the “runica” algorithm was used (EEGLAB [91]) to detect and remove components due to blinking, muscular, and eye movements. Thirty-two (one per electrode) source signals were processed [51,87,88]. An embedded Matlab method (ADJUST) [92] was applied to the EEG signals to discriminate the artifact components by combining stereotyped artifact-specific spatial and temporal features. Certain components out of marked criteria were rejected in the process. A baseline was calculated using a 200 ms segment prior to the target presentation. After, a minimum criterion to maintain an acceptable signal-to-noise ratio was established with 25 artifact-free trials per participant. Due to excessive artifacts, data from 5 participants were excluded from our analysis. For the images in Figure 1, the experimental condition was the product type (ham or turkey) and for the images in Figure 2, the experimental condition was the promotion (with/without the euro sticker). For each participant, the ERP waveforms of each electrode were averaged for all the combinations of the experimental conditions with the priming valence (positive or negative), i.e., eight waveforms per participant and electrode. To study the N400 component, a time window from 400 ms to 600 ms was taken according to similar related studies [88,93,94,95] focusing principally on frontal (F3, Fz, and F4), central (C3, Cz, and C4), and parietal (P3, Pz, and P4) electrodes (see Figure 4).

## 3. Results

Based on the information collected from the electrodes of each subject, a final grand average of ERP waveforms was obtained by combining subjects in each of the experimental conditions mentioned earlier. This final ERP waveform represents the mean amplitude observed in an interval ranging from 0 ms to 600 ms after the onset of the target picture presentation. This grand average reflects overarching patterns among different stimulus conditions. The prominent wave peaks seen in this combined average result from a series of amplitudes present in the individual waveforms produced by each subject due to the methodology employed. The mean activity within the N400 window (400–600 ms) was selected for use in statistical comparisons. This means that N400 activity enabled us to observe the overall effect of priming and the target in the experiment. Two multivariate repeated-measures analyses of variance (ANOVA) were conducted, one for each experimental condition (refer to Figure 1 and Figure 2). The mean N400 activity analysis was performed for each electrode zone, totaling six analyses as dependent variables. The independent factors were valence (positive and negative) and the experimental condition (type of product or promotion). A corrected analysis of sphericity was necessary due to the factors’ conditions. A significance level of 0.05 was used for statistical tests. Figure 5, Figure 6, Figure 7 and Figure 8 display the representation of grand-average ERPs for the two experimental conditions: one for York Ham and Turkey Breast packaging and the other for modified packaging. The results will be analyzed, considering the N400 effect resulting from modulation due to congruence or incongruence. In other words, a perceived congruence or alignment between the priming and the target should result in a smaller negative effect than when there is incongruence between the priming and the target. To enhance this effect, positive targets should be congruent with the packaging, while negative targets should be incongruent.

### 3.1. ERP Evidence for York Ham and Turkey Breast Original Packaging

The results of the first priming–target group corresponding to the original packaging are presented in Table 2. We focus on the parietal electrodes (P3, Pz, and P4), which exhibited significant differences in the analysis. These electrodes are positioned to identify the N400 effect of the ERPs.

The statistical results show a significant difference in the effect ValencexProduct in the P3 position [F(1, 22) = 4.592, *p* = 0.043] and P4 position [F(1, 22) = 13.572, *p* = 0.001]; no significant difference in the Pz position was found (see Table 3). The results of the comparison between the words and the York Ham and Turkey Breast packaging will be characterized by an N400 effect, where the positive valence should exhibit a smaller value compared to the negative valence. This is attributed to the nature of the perception of the packaging and its health content. In this context, congruence should not result in a greater effect than incongruence.

The parietal zone presented a strong activity during the semantic perception process; in this sense, the P3 position will show a significant N400 effect (*p* = 0.043, <0.05). The mean observed for the positive valence of York Ham (MHamP: −1.947 μV) is smaller than for the negative valence (MHamN: −2.172 μV). Meanwhile, the negative valence of Turkey Breast (MTurkeyN: −1.832 μV) was smaller than its positive valence (MTurkeyP: −2.595 μV). For the P4 position, another significant N400 effect (*p* = 0.001, <0.05) was found. The mean revealed that the negative valence of York Ham (MHamN: −2.350 μV) was higher than the positive valence (MHamP: −1.469 μV). Meanwhile, the positive valence of Turkey Breast (MTurkeyP: −2.099 μV) presented a higher amplitude than the negative valence (MTurkeyN: −1.698 μV). The mean amplitudes of the N400 effect in P3, Pz, and P4 are illustrated in Figure 9.

### 3.2. ERP Evidence for modified York Ham Packaging

The results of this analysis are summarized in Table 4, focusing on the same parietal electrodes (P3, Pz, and P4) to identify the same effect in the case of modified packaging. The contrast of the modified packaging did not elicit the desired N400 effect, where it was expected that the positive valence would show an incongruent effect with the worst perceived packaging on the added value of using the euro sticker. However, in this scenario, the negative valence might have a stimulating effect due to a potential connection with the information presented on the modified packaging. The parietal area did not display significant activity during the semantic perception process when exposed to these stimuli. The N400 effect resulting from the priming and target arrangement appears similar for both positive and negative valence.

Nevertheless, some observations can be made based on the mean amplitude results. Specifically, in the case of P3, Pz, and P4, the positive valence shows a higher amplitude when the logo and label are included compared to the same positive valence without this information. This effect is less noticeable for the negative valence in both cases. The mean amplitudes of the N400 effect at P3, Pz, and P4 are illustrated in Figure 10.

## 4. Discussion

The objective of this pilot study was to expose the N400 effect evoked by the contrast of information and packaging artworks. The methodology of semantic priming (adjectives) and target (images) showed evidence that should be highlighted in this discussion. Firstly, we must highlight the EEG technology used to obtain the ERP recordings. This technique and its methodology provided evidence that the N400 effect recordings can be obtained in this study type we conducted. Although this technique initially had a clinical purpose [42,43], we can affirm that its adaptation [45,46,48] is in accordance with published experiments. The most important result was the confirmation that the parietal region showed the expected effect, showing the negative peak corresponding to the N400 effect evoked by priming and targets. This result indicates that we are working with information about the cognitive processing corresponding to the semantic load of the adjectives and how it can establish a connection with the observed products. Our results are in alignment with previous research, which has indicated that the N400 is sensitive to semantic processing in incongruent compared to congruent conditions [70,71,72,73,89,96].

Delving further into the study’s findings, it has been corroborated that the N400 effect can serve as an electrophysiological measure that provides objective information to measure congruent or incongruent relationships; however, it should be mentioned that with relevant technical aspects during the process of matching adjectives with container images, the N400 effect becomes evident. The variation within the time window from 400 ms to 600 ms exhibited a similar effect as seen in related studies where the emergence of this effect was proposed [88,93,94,95,97]. However, when considering studies more closely related to packaging, the observation window occurs roughly 50 ms earlier [36,84,86,87,97]. This may imply an important adjustment for the experiment, as the absence of a significant difference in all components could potentially be attributed to this time window choice.

Continuing with the findings, the statistical results unveiled that activity in the parietal area responds to the N400 effect, induced by the congruence or incongruence contrast that arises when the adjectives and images align or do not align with the participants’ perception. The desired ERP effects were observed, especially in the cases where the most pronounced negative variation between priming and target could be observed. In the P3 position case, the contrast result with the original packaging, incongruence evidence was found between positive valence and the Turkey Breast packaging. There was also incongruence evidence between negative valence and the York Ham packaging. In this scenario, the measurement indicates that the congruent perception of the participants is a negative association for the Turkey Breast packaging and a positive association with the York Ham packaging. Similarly, in the P4 position case, the contrast result with the original packaging, there was incongruence evidence between the negative valence and the York Ham packaging and incongruence evidence between the positive valence and the Turkey Breast packaging. This finding reaffirms what was found for P3, showing how the parietal area was active during the cognitive process of this pilot study.

These findings broadly support the work of other studies [81,82,84,97,98], and those negative adjectives indicate a conflict about packaging type that participants consider positive. However, in our research, this resolution was shown as a positive appraisal of the York Ham packaging and a negative appraisal of the Turkey Breast packaging. The second part of the experiment with the modified packaging did not yield conclusive information that we could contrast with the findings of the first part. However, we can possibly intuit the reasons why the artwork did not have such a marked effect.

Taken together, these findings suggest that the role of negative adjectives has a much greater impact in detecting incongruence in participants’ perceptions. In our study, the positive and negative adjectives elicited from the attributes and features related to the brands and the product served their purpose. Likewise, original packaging images and their two categories were good targets for a contrasting evaluation, marking evidence of the preference for one of the two products, in this case, the York Ham packaging. In our study, we did not have an antecedent for product/food preference by the participant, so the reason behind this choice will be left for future studies, as it was not the purpose of the research. However, when using the York Ham packaging for the second study, the evaluation was not contrasting, giving evidence that these modifications did not influence the global perception.

## 5. Limitations and Future Perspectives

This study has contributed several findings to ERPs’ use in this type of research. Nonetheless, limitations and areas of opportunity for future research should be considered. The first consideration is the semantic priming choice. In our case, the selection was based on taking information regarding the attributes given by the brand. However, previous exercises could be worked on to generate words with a purpose more adjusted to the research and provoke or trigger a greater N400 effect. The second consideration is the treatment or selection of the visual elements as targets. In our research, packaging includes a wide variety of stimuli (such as branding, logos, text, information, labels, etc.) that may represent potential triggers. However, it can be difficult to isolate precisely which of these elements is responsible for triggering the effect. Therefore, the recommendation is to have a traditional exercise beforehand to contrast results. The final consideration is a more appropriate selection of the sample or participants. In our case, the participants were university students, who may have a biased perception of the product to be evaluated. It is essential to have traditional questionnaires or questionnaires that provide more information on the tastes and preferences of the participants.

## 6. Conclusions

In this pilot study, we have made a significant contribution to the ERP technique in gaining a deeper understanding of the cognitive impact and processing related to food packaging. An objective measurement to understand perception has allowed us to unveil new aspects of design packaging related to consumer preference. The observed N400 effect increases the value of employing and exploring this measure as a compelling addition to traditional research, linking the neuroscientific perspective. Finally, the proposed methodology can serve as a draft for future studies aimed at investigating the cognitive and emotional processes associated with food packaging. However, as far as we are concerned, there is still a research gap in the packaging area from a neuroscience perspective.

## Figures and Tables

**Figure 1 foods-13-01876-f001:**
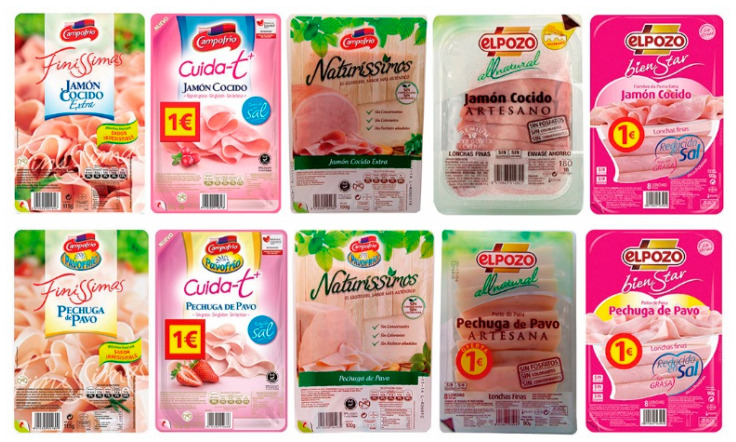
York Ham packaging (**top**) and Turkey Breast packaging (**down**) of Group 1.

**Figure 2 foods-13-01876-f002:**
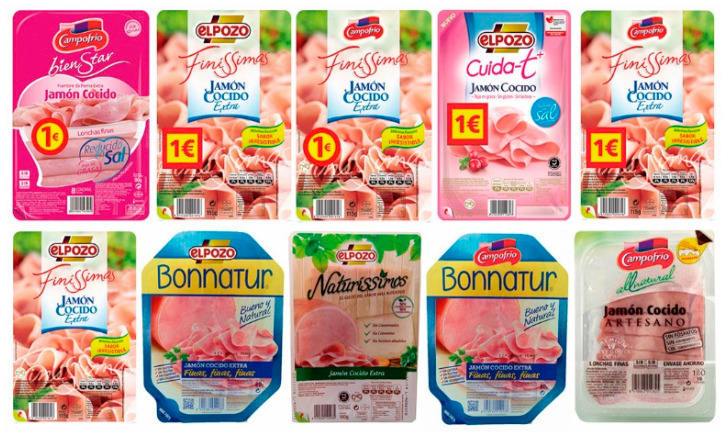
Modified packaging of York Ham with euro stickers and logos (**top**) and without euro stickers (**down**) of Group 2.

**Figure 3 foods-13-01876-f003:**
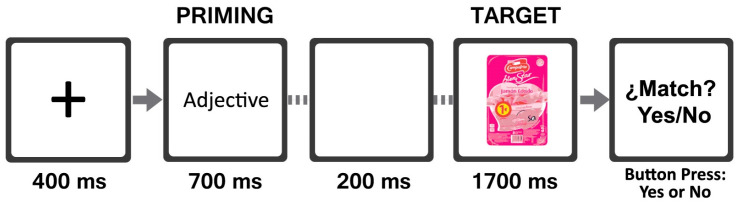
Trial construction and timeline.

**Figure 4 foods-13-01876-f004:**
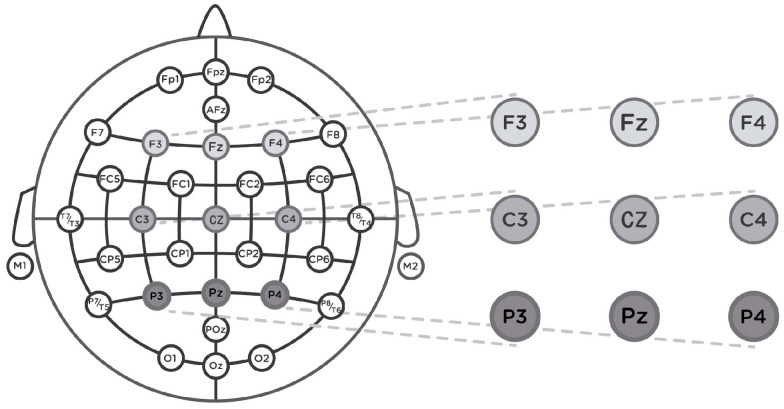
International 32 electrode placement system, and the nine electrodes used in this study.

**Figure 5 foods-13-01876-f005:**
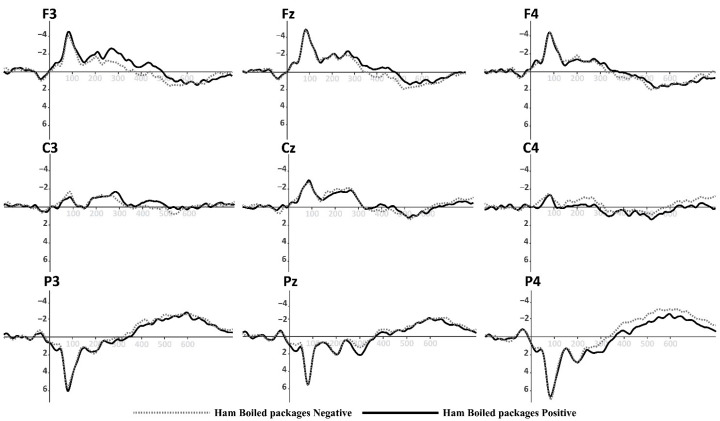
Grand-average ERPs for York Ham in original packaging from Figure 1 (**top**). Black line for positive primes, dotted line for negative primes.

**Figure 6 foods-13-01876-f006:**
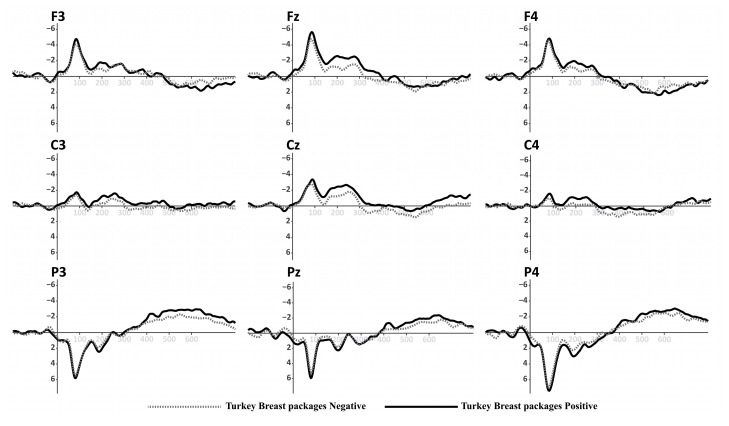
Grand-average ERPs for Turkey Breast in original packaging from Figure 1 (**down**). Black line for positive primes, dotted line for negative primes.

**Figure 7 foods-13-01876-f007:**
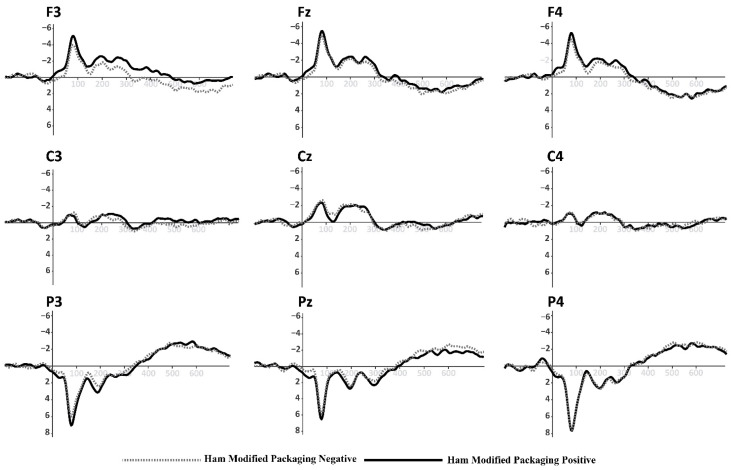
Grand-average ERPs for York Ham in modified packaging with euro sticker and logos added from Figure 2 (**top**). Black line for positive primes, dotted line for negative primes.

**Figure 8 foods-13-01876-f008:**
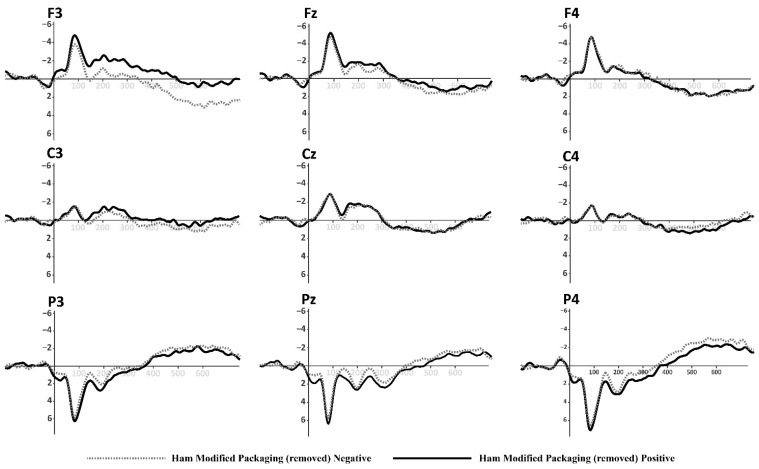
Grand-average ERPs for York Ham in modified packaging with euro label and logos removed from Figure 2 (**down**). Black line for positive primes, dotted line for negative primes.

**Figure 9 foods-13-01876-f009:**
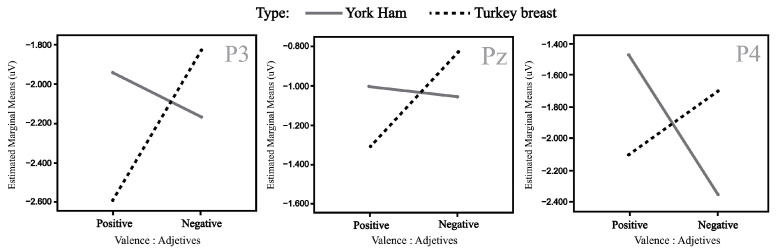
Mean amplitude (µv) for significant P3 (**left**), Pz (**center**), and P4 (**right**) electrodes in the interaction type and valence.

**Figure 10 foods-13-01876-f010:**
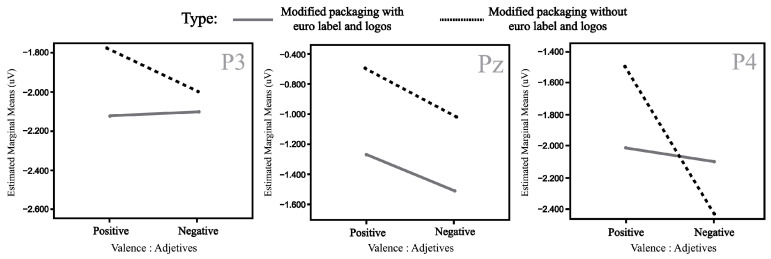
Mean amplitude (µv) for significant P3 (**left**), Pz (**center**), and P4 (**right**) electrodes in the interaction type and valence (modified packaging).

**Table 1 foods-13-01876-t001:** Positive and negative adjectives are used as prime words.

Positive Adjectives	Negative Adjectives
Atrayente/Attractive	Común/Common
Barato/Cheap	Caro/Expensive
Delicioso/Delicious	Repugnante/Disgusting
Nutritivo/Nutritious	Insulso/Dull
Saludable/Healthy	Nocivo/Unhealthy

**Table 2 foods-13-01876-t002:** Means for P3, Pz, and P4 electrodes for York Ham and Turkey Breast packaging.

P3 Electrode			Pz Electrode			P4 Electrode		
Valence	Packaging	Mean (µv)	Valence	Packaging	Mean (µv)	Valence	Packaging	Mean (µv)
Positive	York Ham	−1.947	Positive	York Ham	−1.010	Positive	York Ham	−1.469
	Turkey Breast	−2.595		Turkey Breast	−1.370		Turkey breast	−2.099
Negative	York Ham	−2.172	Negative	York Ham	−1.159	Negative	York Ham	−2.350
	Turkey Breast	−1.832		Turkey Breast	−0.865		Turkey Breast	−1.698

**Table 3 foods-13-01876-t003:** Multivariate tests for P3, Pz, and P4 electrodes for York Ham in original packaging.

Electrode	Effect	Value	F	GI Error	Sig.
P3	Valence	0.966	0.767 a	22.000	0.391
	Product	0.983	0.378 a	22.000	0.545
	ValencexProduct	0.827	4.592 a	22.000	0.043
PZ	Valence	0.981	0.432 a	22.000	0.518
	Product	0.999	0.013 a	22.000	0.911
	ValencexProduct	0.914	2.062 a	22.000	0.165
P4	Valence	0.970	0.670 a	22.000	0.422
	Product	1.000	0.003 a	22.000	0.959
	ValencexProduct	0.618	13.572 a	22.000	0.001

a—Exact Statistic.

**Table 4 foods-13-01876-t004:** Means for P3, Pz, and P4 electrodes for modified York Ham packaging.

P3 Electrode			Pz Electrode			P4 Electrode		
Valence	Modified Packaging	Mean (µv)	Valence	Modified Packaging	Mean (µv)	Valence	Modified Packaging	Mean (µv)
Positive	Sticker-Logo	−1.947	Positive	Sticker-Logo	−1.010	Positive	Sticker-Logo	−1.469
	No sticker-Logo	−2.595		No sticker-Logo	−1.370		No sticker-Logo	−2.099
Negative	Sticker-Logo	−2.172	Negative	Sticker-Logo	−1.159	Negative	Sticker-Logo	−2.350
	No sticker-Logo	−1.832		No sticker-Logo	−0.865		No sticker-Logo	−1.698

## Data Availability

The data used to support the findings of this study can be made available by the corresponding author upon request.

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
