# Peer review of "Using Event-Related Potentials to Evidence the Visual and Semantic Impact: A Pilot Study with N400 Effect and Food Packaging"

_foods, 2024, doi:10.3390/foods13121876_

Round 1
Reviewer 1 Report
Comments and Suggestions for Authors
It is an interesting work. A major drawback of this work that cannot be avoided is the uncertainty caused by individual differences and experimental conditions. However, the authors only selected a very small number of non-representative samples, which cannot effectively solve these problems. Anyway, the work still might be considered after undergoing major revisions.
1. Introduction should provide more descriptions to help to understand this method. Also, more previous studies are also beneficial for the introduction.
2. L147: The sample size is so small. How to verify the reliability of the conclusion?
3. The authors only selected undergraduate students with a very narrow age range participated in this work. It cannot represent the overall consumer population. Meanwhile, the authors did not specifically mention the uniqueness of the sample in the title, abstract, discussion, and conclusion, etc.
4. Stimuli's choice seems too arbitrary and subjective, and it should provide more evidence to prove the rationality of this experimental design.
5. Similarly, it should provide more evidence to prove the rationality of the design of positive and negative adjectives as a prime word.
6. L227: Fig. 4 is unclear.
7. Fig. 5~8 seem to be results, not methods.
8. The discussion section is too general and requires a more in-depth analysis of the experimental results.
Comments on the Quality of English LanguageModerate editing of English language required.
Author Response
The authors would like to express our heartfelt gratitude to the reviewers for their invaluable comments and suggestions. Your contributions have significantly enhanced the quality of our manuscript. The original comments of the reviewer are in red color.
- Introduction should provide more descriptions to help to understand this method. Also, more previous studies are also beneficial for the introduction.
The content of the introduction has been restructured. You will find two subtopics that will help you to better understand the purpose of our research. The second topic, which has to do with the technique used in the research, was further developed using reference literature, emphasizing technical aspects marked by similar and/or previous studies.
- L147: The sample size is so small. How to verify the reliability of the conclusion?
We have applied a sample size above the mean value typically used in this type of study. Šoškić et al. (2022) analyzed 132 N400 studies, observing a mean N of 20.2 participants. This information has been added to the text of the manuscript.
Šoškić, A., Jovanović, V., Styles, S.J. et al. How to do Better N400 Studies: Reproducibility, Consistency and Adherence to Research Standards in the Existing Literature. Neuropsychol Rev 32, 577–600 (2022). https://doi.org/10.1007/s11065-021-09513-4
- The authors only selected undergraduate students with a very narrow age range participated in this work. It cannot represent the overall consumer population. Meanwhile, the authors did not specifically mention the uniqueness of the sample in the title, abstract, discussion, and conclusion, etc.
After making improvements to the manuscript, we have taken your suggestions into account. The first modification is the title of the paper, making it clear that it is a pilot study that seeks to apply the ERP technique in the context of food packaging. The second modification is in the limitations and future work, making it clear that it was one of our restrictions in this pilot study.
- Stimuli's choice seems too arbitrary and subjective, and it should provide more evidence to prove the rationality of this experimental design.
The selection of the stimuli was a decision based on the interest of the line of research. Previous eye-tracking research has been conducted for other types of packaging. Exploring this type of packaging contributes to an area of opportunity since, to date, the number of research studies that contrast this type of technique with food packaging is still limited.
- Similarly, it should provide more evidence to prove the rationality of the design of positive and negative adjectives as a prime word.
The selection of words for our study is influenced by the dynamics of reference studies. The rationale behind the selection of semantically valence words (positive and negative) is to elicit the effect of congruence or incongruence. In our case, due to the nature of the containers, positive attributes were selected; therefore, the negative charge should be their opposite. However, in the study's limitations, it is described that previously there could be an exercise to validate this selection.
- L227: Fig. 4 is unclear
The image was modified to highlight the positioning of the six chosen electrodes.
- Fig. 5~8 seem to be results, not methods
The images were placed in the recommended section.
- The discussion section is too general and requires a more in-depth analysis of the experimental results.
After the suggested improvements, the discussion was also restructured to emphasize several findings and contributions of this work. We welcome your feedback on any new details in this new manuscript version.

Reviewer 2 Report
Comments and Suggestions for Authors
1. Check the punctuation marks in the abstract. Line 23.
2. In abstract and conclusion, please refine the experimental results in the paper respectively.
3. Please modify the Introduction part to emphasize the significance of this study.
The work of this paper is meaningful, and it is suggested that the author should clarify the structure and conclusion of the paper more clearly.
Author Response
The authors would like to express our heartfelt gratitude to the reviewers for their invaluable comments and suggestions. Your contributions have significantly enhanced the quality of our manuscript. The original comments of the reviewer are in red color.
- Check the punctuation marks in the abstract. Line 23
The abstract section has been improved.
- In abstract and conclusion, please refine the experimental results in the paper respectively
The abstract content has been improved after reviewing all the reviewers' comments. The text in the abstract has been improved to better present the experimental content of the paper. Also, the conclusions content has been refined based on the findings of this experimentation. We are open to receiving further feedback on these two sections of the manuscript.
- Please modify the Introduction part to emphasize the significance of this study.
After the suggested improvements, the discussion and conclusion section was also restructured to emphasize several findings and contributions of this work. We welcome your feedback on any new details in this new version of the manuscript.

Reviewer 3 Report
Comments and Suggestions for Authors
t is suggested that the authors also highlight the consumer response to the information to be included in the packaging according to the regulations (nutrition label etc.).
It is suggested to reduce the introduction to make it more readable.
The number of consumers is somewhat reduced as they justify a limited number of consumers
Comments on the Quality of English LanguageThe quality of Englishe language is appropriate
Author Response
The authors would like to express our heartfelt gratitude to the reviewers for their invaluable comments and suggestions. Your contributions have significantly enhanced the quality of our manuscript. The original comments of the reviewer are in red color.
- It is suggested that the authors also highlight the consumer response to the information to be included in the packaging according to the regulations (nutrition label etc.).
The objective of the pilot study is not related to the perception of nutritional attributes or to the observation of nutritional information on packages. Therefore, there is no information on this subject in this study. However, in our limitations, we touched on the issue of validating word selection based on preliminary exercises. One of these could be related to what you suggest. We appreciate your observation.
- It is suggested to reduce the introduction to make it more readable.
The introduction content has been restructured. You will find two subtopics that will help you better understand our research’s purpose. The first subtopic " Seeking Objective Evaluation of the Impact of Packaging Design" shows the relevance of the perception of food packaging, analyzing the visual and written impact on consumers to improve the design, using innovative techniques. The second subtopic is " Understanding Cognitive Processes in Consumer Perception Through Event-Related Potentials and N400 Effects,” which shows the use of electroencephalogram (EEG) and event-related potential (ERP) N400 reflect the congruence or incongruence between words and images on product packaging, influencing consumer perception. We welcome your feedback on any new details in this new version of the manuscript.
- The number of consumers is somewhat reduced as they justify a limited number of consumers.
We have applied a sample size above the mean value typically used in this type of study. Šoškić et al. (2022) analyzed 132 N400 studies, observing a mean N of 20.2 participants. This information has been added to the text of the manuscript.
Šoškić, A., Jovanović, V., Styles, S.J. et al. How to do Better N400 Studies: Reproducibility, Consistency and Adherence to Research Standards in the Existing Literature. Neuropsychol Rev 32, 577–600 (2022). https://doi.org/10.1007/s11065-021-09513-4

Round 2
Reviewer 1 Report
Comments and Suggestions for Authors
The authors have made some modifications. It seems that the manuscript could be accepted for publication in its current form.
Comments on the Quality of English LanguageMinor editing of English language required.